# National debt management and business sustainability in Africa's largest economy: A focus on the private sector

**Mengya Shang[1], Uchechukwu E. Okorie[2], Yin Hang[1], Xiaosong Jin[3]\*, Daniel E. Ufua[4]\***

**1** School of Economics and Management, Harbin Engineering University, Harbin, 150001, China,
**2** Department of Economics and Development Studies, Covenant University, Ota, 112104, Nigeria,
**3** Development Strategy Research Office, Northeast Agricultural University, Harbin, 150028, China,
**4** Department of Business Management, Covenant University, Ota, 112104, Nigeria

\* jinxiaosong@neau.edu.cn (XJ); daniufua@gmail.com (DEU)

## Abstract

In many developing economies, high and increasing public debt profile constitutes an essential means of financial risk. An appropriate debt management is germane for survival of business and good international reputation though its effect on private sector credit mobilization had been seldomly investigated. This study seeks to know whether strategic debt management approach exacts a significant effect on the Nigerian private sector and Africa at large resulting to higher credit availability for sustainable enterprise establishment. The study used a time-series observation spanning from 1981–2021. The method of data analysis employed the unit root test for stationarity. Johansen cointegration and vector error correction approach. The result of the unit root test indicates the series were all stationary after first difference and thus were integrated of order1. The Johansen cointegration test support the existence of a cointegrating series between the private credit and its determinants. More empirical evidence from the study shows that proper debt management and increase revenue generation through net taxes on products accounted for 0.93 and 1.32% increase in private sector credit mobilization, while total external debt stock was responsible for a significant negative influence of 0.60% on private sector credit mobilization. The study recommends that the government should always be proactive in their strategic and innovative approach to debt management, revenue generation and sources of funds. This will help not only to avoid crowding out of the private sector but will enhance adequate credit mobilization for effective operations of the private sector.

## 1. Introduction

The management of national debt has become a common practice among economies around the world. Many national economies borrow to either supplement the funding for their budget or execute special projects in their economies [1,2]. The use of debt funding has helped the debtor economies to support the provision of basic amenities required to sustain the lives of the citizens and business activities within the economy. National debts are also adopted as an

**Data Availability Statement:** All relevant data are within the manuscript and its Supporting Information files.

**Competing interests:** The authors have declared that no competing interests exist.

option to address macroeconomic crises in economies and an available option to avoid an increase in the tax burdens on the citizenry [3,4]. While many economies have gone into borrowing on a higher scale especially when the interest and payback are attractive, coupled with their needs to meet national economic needs, the burden of debt repayment requires the continuous obligatory actions of the debtor nation to affect the payback as negotiated [5].

Business organizations in an economy depend on the fluent interactions and movement of basic macro support such as the availability of credit facilities, and infrastructural provisions to function in an economy. This means that these businesses that are largely resident in the private sector are expected to participate in the payback of acquired national debt through their remittance of taxes and levies to the government treasury [6–10]. The management of national debt and its influence on business sustainability has become a trendy practice in Nigeria, which leaves the government with the responsibility to determine the amount and rate at which the national debt is acquired and the direction of such funding to support economic activities, with the aim to sustain businesses in Nigeria [11–13].

The current study is focused on exploring the impact of the Nigerian national debt profile on business sustainability in the private sector of the economy. It aims to provide learning about the level of support accessible to operating businesses in the private sector due to the acquisition of debt by the Nigerian national government. It is focused on developing learning on the intrinsic implications of national debt burdens on the operations of businesses in the private sector (see [2,14]. A critical question raised in this study is *how can the use of national debt funding provide further support to business activities in the various sectors*? and *how in turn, can the extant productivities of the private sector be harnessed to support national debt management obligations in Nigeria*? This study aims to address these critical questions and suggest an improved approach to maximize the positive impacts of national debt burdens on the sustenance of businesses in the Nigerian economy.

While this study explores the potency of the national debt in supporting business activities, culminating in investment drives in the private sector, its emphasis is on the need for prudent management and allocation of national to achieve this set purpose, in order to maximize the useful of national debt to the broad economic development such as positive transformation of the private sector. The study therefore emphasizes the viability of national debt management, in terms of its sustainability in critical areas such as the development of a robust repayment structure that offers effective support to the sustenance of business activities in the economy. This aligns with the observation of [15], who based their study on the natural resource management approaches of selected developing economies, towards attaining economic growth and the optimization of financial productivity for optimum impact on the economies.

This study assumes the following structure: The next session provides a detailed literature review on the key concepts of national debt management and business sustainability and the underpinning theory adopted in this study. This is followed by the methodology and a discussion of the main findings. The final section covers the conclusion and recommendations for further research.

## 2. Literature review

The management of national debt has become essential to private sector development. This is as a result of the inadequate funding to meet most national budgets at a point in time. Debt management is the entire ambiance of government policies on the acquisition disbursement and repayment of a debt by the national government [16,17]. National debt assumes various classifications, ranging from local, in which case, the debts are sourced from creditors within the economy from sources such as resident banks syndications, sale of government bonds, and related bills. These local national debts are usually held in the local currency.

The local debt in Nigeria is usually in short or medium terms, ranging from months to years. The other form is external debt whereby a national government can source debt funding from external creditor organizations such as the IMF, Paris Club, and other national government or their agencies [18,19]. These are debt facilities acquired to finance identified economic developmental issues of interest. Such can also include the need to provide extant support for business operations in the private sector. For instance, the Nigerian government issued SUKUK bonds to raise funding to finance special projects in the economy such as road construction for effective business and humanitarian activities [20].

According to [21], macroeconomic projection such as national debt management can enhance financial development in the direction of environmental quality, through the engagement of the participating stakeholders such as investors and practicing business entities, in a process of critical innovation that seek to maximize the available macroeconomic resource such as national facilities, with an embedded nexus for environmental preservations. This observation, therefore, points out that the failure of the Nigerian public debt management agencies to achieve the objective of supporting the private sector with acquired public debt can result in chaotic situations such as breaches in the national debt service obligation to the creditors, poor credit rating, negative environmental impacts, and gross macroeconomic performances. And many private sector businesses can, therefore, be faced with challenges such as lack of access to funds for working capital, general business insolvency, and ultimately failure [22].

The Nigerian private sector is broadly divided into two parts, namely the organized private sector and the informal sector. The Nigerian public debt management agency takes the responsibility to ensure that the private sector benefits from national debt management activities in Nigeria. The Nigerian private sector has remained a hub for socio-economic shocks absorbent that creates the balance and sustainability of the national economy, through activities such as business relationships and transactions that shape the directions of the national economy [23].

The organized private sector has regular business operations that are duly recognized by the Nigerian economic system. These include organizations such as commercial banks, insurance companies, and organized private health services organizations. [24] reckon that national debt has a significant impact on economic stability in the long run. This implies that the government, being the umpire stakeholder, both in national debt management and business support services, has the responsibility to embrace long-term policy-making activities that can offer the needed leverage for better economic results through effective management of national debt and the strive to support business sustainability in the Nigerian private sector [see 25].

The informal sector comprises the unaccounted or sundry sector that hosts several businesses that mostly operate at a micro-level. They are usually unregistered and yet contribute to national economic development. While the Nigerian government strives to support private sector businesses through activities such as extant microeconomic policies and debt management practices, the need for further effort to support the transformation of informal sector businesses has remained an obvious challenge [26,27].

Researchers and macroeconomic observers reckon that most governments, including Nigeria, focused in this study engage in the continuous strive to offer support to the transformation of the informal sector to a formal status so as to attain the sustainability and growth of such businesses operating in this sector of the economy [28,29]. They argue that through their transformation they are positioned to make better contributions to the national economy. As a result of this, it has been observed that the Nigerian government has on several occasions provided soft loans to finance business transformation and sustainability [30,31].

Elsewhere, [32], studied economic growth and financial development in the Greek private sector, suggesting the need for policymakers to focus on addressing the challenging

environmental effects of business activities in the Greek tourism sector. This substantiates the focus of this study, on how the umpire stakeholder can enhance national debt management for maximum impact on socio-economic development across sub-sectors in the Nigerian economy [25,33]. Similarly, [34], in their study in the Chinese background, note that extant use of financial support to business activities such as investment in green economic management can facilitate a productive medium to engage participants in the critical economic sectors, to create a high-quality economic development, which can in turn, eliminate the mismatch between the expectation of the stakeholders and the real-life trend of advancement across sector of the economy.

In another study on the Ghanaian economy, [22] highlights certain challenges against effective national debt management practices and economic support for the private sector. These include the temptation to use business loans for acquiring capital items, tax evasion, poor revenue recording systems, the unfair trade practices from dumped Chinese goods. He suggests the need for productive engagement of key stakeholders by the national government to develop the necessary cooperation to ensure successful national debt management practices and extant support for the private sector. Based on the current review narrative, Fig 1 below presents a schematic model representation of national debt management activities and their effects on business sustainability in Nigeria.

Furthermore, as represented in Fig 1, there is conspicuous connectivity between national debt management and private sector support in Nigeria. Nevertheless, what remains a critical

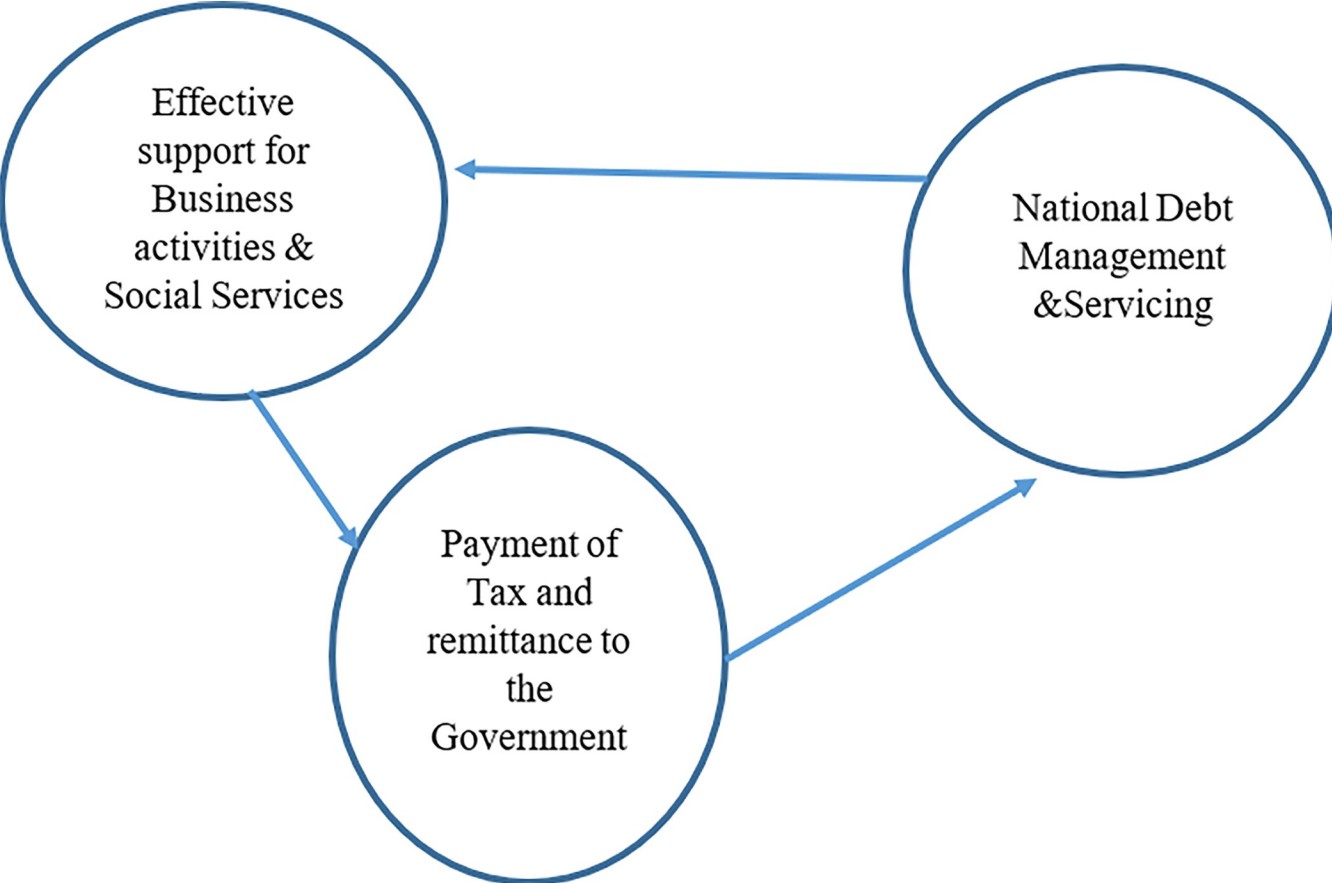

**Fig 1. The schematic model of national debt management's effects on business activities in Nigeria.** Source: Author's compilation.

focus in this study is how can such existing connectivity be harnessed for a positive contribution to the national economy, which can in turn, solidify the national credit rating to attract more debt facilities to continuously address key national issues such as the sustainability of the private sector. Fig 1 depicts the productivity growth effects of effective management of national debt. These includes effective support for business, investment and social services in the economy. As shown in Fig 1, it also spurs the willingness of compliance among stakeholders such as business owners in their response to remittances and tax payments to the government treasuries, which in turn results in the seamless servicing of acquired national debt and improvement of nation rating. [24,35].

### 2.1 Underpinning theory

This study is anchored on systems theory. It draws on the structuring of the national debt profile, the alignment with the operations of the Nigerian private sector the purpose of providing financial support to the functioning of the private sector, and the contribution to national economic development [36]. The justification for the adoption of systems theory is based on its embedded focus on the end-to-end effects of management practices [37].

In this study, systems theory focuses on the impacts of national debt, the engagement of relevant stakeholders such as financial intermediaries, and extant government decision trends regarding the support of business activities through prompt management of the national debt profile, which can enhance their contribution to the growth and development of the national economy. Based on systems theory, this study is therefore anchored on the effective management of the connectivity of the key variables and the stakeholders in the engagement of national debt and business and social economic advancement drive in the Nigerian private sector [see 38]. The next section presents the details of the methodological approach adopted in this study.

## 3. Materials and methods

This study focused on the effect of national debt management on business sustainability with emphasis on private sector enterprises in Nigeria. Secondary data from Central Bank of Nigerian [39,40] were utilized in the course of the research. Given the high stochastic nature of the time series observations characterized by random walk, it becomes imperative that the stationarity of the data employed by this study should be ascertained before proceeding for further analysis. In this study, the stationarity test was tested at level and first difference.

The first difference becomes necessary after the observations were seen to be non-stationary at their levels. The non-stationarity of the series thus depicts the presence of unit root in the series. This could give rise to spurious results when the series are estimated at this stage, hence the need for the data to be differenced to attain to a stationary series. All the observations were differenced once to achieve stationarity which suggests the data were all integrated of order 1. The integration of the series is an indication of a cointegrated series signifying a long-run relationship among the variables. Hence, the study proceeded to employ the [41,42] cointegration approach in Eq (1) to examine whether there is an existence of cointegration among the variables in the model using the Trace and Maximum Eigen value statistic. In this approach, the calculated test statistic is compared with its critical value at 5 percent significance level. A greater test statistic in relation to the critical value connotes the existence a cointegrated series.

In the present study, the Trace and Maximum Eigen statistic revealed at least one cointegrated series. The Johansen cointegration procedure does not only answer the question on the existence of a cointegrated series but also indicates the number of cointegrated equation(s) as well as useful information about the cointegrating coefficients in the model. The Johansen

methodology is based on the vector autoregression (VAR) of order p expressed as;

$$Y_t = \alpha + A_1 Y_{t-1} + \ldots + A_p Y_{t-p} + £_t \qquad (1)$$

Where;

$Y_t$ is nx1 vector of variables that are integrated of order1 represented as I(1) and

$£_t$ is nx1 vector of innovations.

$A_1 \ldots A_p$ connotes the autoregressive coefficients in the VAR model.

$Y_{t-1} \ldots Y_{t-p}$ denotes lag order 1 to order 'p' in the autoregressive estimates

The study went further to ascertain the short-run dynamic process that determines the long-run state and the mechanism of systemic adjustment through the Vector Error Correction model (VECM) approach. In the VECM procedure the long-run and short-run estimates were ascertained with the model adjustment mechanism through the error correction term associated with the estimated model. The relevance of the model dynamic adjustment process was evaluated from the view point of the error correction term obtained within the unit circle, level of statistical significance and the associated negatively signed coefficient supporting the convergence of the system in regard to exogenous shocks on the system.

The Vector Error Correction Model (VECM) is a statistical model used to analyze the relationship between multiple time series variables. It is mostly employed when the variables in the specified model are cointegrated. This implies that the variables exhibit a long-run relationship. The VECM technique is an extended Vector Autoregressive Model (VAR) that is particularly suitable for integrated multivariate time series observations. The most prominent advantage of this technique over the VAR model is that it accounts for the short-term and long-term dynamics between variables of interest. The technique does this by the incorporation of an error correction term that accounts for the adjustment mechanism toward the long-run equilibrium. This makes this method specifically essential for the analysis of economic and financial time series data characterized by the variable exhibition of fluctuations in short-term with long-term trends.

Also, the VECM is advantageous in handling the issues of spurious regression that can be incurred using the VAR model to integrate time series. This refers to a scenario where the T-statistics of the estimates are highly significant and R-squared is high amidst no meaningful relationship between the variables. This bias is controlled by VECM modeling the co-integrating relationship between the variables, thus ensuring that any observe relationship is not by coincidence.

In term of the study design, this study opted for vector error correction approach having ascertain the theoretical justification for the existing of a long-run relationship between private sector credit mobilization and debt management approach through the unit root stationarity procedure. The VECM result also provide valuable insights into the dynamics of the variables and help identify any long-term relationships that may exist. Notably the choice of the study design and method was based on the management of potential bias from the estimation of variables of the same order of integration. It also entails a careful consideration of the underlying theory and empirical evidence in support of the existence of a cointegrating relationship among variables of interest in the present study. In handling potential bias from the study design and method, the assumptions underlying the VECM in terms of the absence of serial correlation and heteroskedasticity of the residual series were met.

## 3.1 Model specification and variable selection

The model specification is strongly linked to the schematic model of national debt management (Fig 1) and the system theory. It is expected that improvement in national debt

management and control measures of external debt stock will trigger a corresponding positive effect on private sector capital flows as a result increased investors' confidence in the national economy. This will invariably enhance the effective credit support system for business transactions and the social cum economic development of the country. In the presence of adequate capital inflows for private sectors activated through the systemic intermediation role of financial institutions and government policies investment will consequently drive. With increases in investment and economic boom, government revenue is increased through payment of taxes and remittance to the government which will in turn boost the provision of public goods and creation of more enabling environment for businesses to drive and more financial deepening in the economy. Consequently, there is less crowding out of the private sector by the government in the process of sourcing capital for business growth.

In this study a linear model has been specified to captured the functional relationship between private sector credit and public debt management in Nigeria as shown in Eq 2. The justification for variables selection derives from the earlier study on private sector credit model of [43] as a function of sovereign debt crisis. Their study argues that with the sovereign debt effect on aggregate demand and country risk premia, restructuring it could adversely influence the private sector's access to foreign capital markets. The present study further argues that public debt management approach could exact a significant effect on private sector mobilization for particularly developing African economies like Nigeria. The model is given in Eq (3)

$$LPSC = f(LPDM, LNTP, LTEDS) \tag{2}$$

Eq (1) is expressed in its explicit function as

$$LPSC = \beta_0 + \beta_1 LPDM + \beta_2 LTNP + \beta_3 LTEDS + \mu_t \tag{3}$$

Where LPSC means Private Sector Credit Mobilization, LPDM stands for Public Debt Management, LNTP represents Net Taxes on Product, LTEDS captures Total External Debt Stock.

The estimated parameter consists of $\beta_1 \ldots \beta_3$ while, $\mu_t$ captures the disturbance term in the model

## 4. Results and discussion

This section presents the organization of the result and its discussions. The study first conducted the unit root test to ascertain the stationarity of the time series observation (Table 1), then the Johansen cointegration test was employed to determine whether there exists a long-

**Table 1. Unit root result.**

| Variable | ADF Test @Levels (5% significance level | ADF Test @ First Difference | Remark |
|---|---|---|---|
| LPSC | -.0.910671 (-2. 936942) | -4,522652 (-2.938987) *** | Integrated to order 1 |
| LPDM | -1.507748 (-2. 938987) | -4.699629 (-2.938987) *** | Integrated to order 1 |
| LNTP | 0.431009 (-2. 936942) | -7.398274 (-2.938987) *** | Integrated to order 1 |
| TEDS | -2.743458 (-2.936942) | -4.868602 (-2.938987) *** | Integrated to order 1 |

Source: Authors compilation.

Note; the critical T-values are in brackets and computed ADF test statistics are outside the bracket

***, indicates stationarity at 1%.

**Table 2. Co integration analysis.**

| Hypothesized No. of CE(s) | Eigen Value | Trace Statistics | 0.05 Critical Value | Prob.** | Max-Eigen Statistic | 0.05 Critical Value | Prob.** |
|---|---|---|---|---|---|---|---|
| None * | 0.554132 | 52.29205 | 47.85613 | 0.0181 | 31.50157 | 27.58434 | 0.0149 |
| At most 1 | 0.289991 | 20.79048 | 29.79707 | 0.3708 | 13.35664 | 21.13162 | 0.4199 |
| At most 2 | 0.153609 | 7.433835 | 15.49471 | 0.5278 | 6.504200 | 14.26460 | 0.5494 |
| At most 3 | 0.023555 | 0.929635 | 3.841466 | 0.3350 | 0.929635 | 3.841466 | 0.3350 |

Source: Authors compilation.

run relationship among the variables which are all integrated of order 1 (Table 2). The vector autoregression lag order selection criteria were conducted to determine the lag length of the variables using Aikaic Information Criteria (AIC) in Table 3. Having conducted the preliminary tests to ascertain the stochastic nature of the time series, the study further was able to estimate the long-run coefficients of the private sector credit model using the normalized long-run estimates of the cointegrating equation (Table 4).

The long-run estimates provide the empirical evidence for the study research question that attempts to investigate the extent public debt management could influence private sector credit inflows for developing African economies like Nigeria. Finally, the vector error correction procedure in Table 5 was used to examine the systemic adjustment of the model in the presence exogenous shocks. It also provides insight on the system nature of convergence or otherwise.

This section thus, presents the result of the stationarity test conducted in the study using the augmented dickey fuller (ADF) test, Johansen cointegration test for evaluating the existence of a long-run relationship among the variables in the model and the Vector Error Correction Model (VECM) that helps to estimate the dynamic long-run and short-run parameters of the model. It further offers the opportunity of assessing the model adjustment process from the short-run dynamic to the long-run state. These results were presented and discussed in the succeeding sections.

The unit root result (Table 1) was conducted to determine the level of stationarity of the variables employed by the model using the Augmented Dickey Fuller (ADF) test. The rationale for this test was informed by the fact that most time series observation are usually characterized by high stochastic trends that influences their stationarity and reliability of their estimates. Variables of the model tested were private sector credit performance (PSCP) as the dependent variable. The independent variables consist of exchange rate (EXR), public debt service (PDS), public debt management (PDM), Net taxes on product (NTP) and total external debt service ratio (TEDS). The result shows that none of the variables was stationary at level, hence all the variables were subjected to second differencing to achieve a stationary trend as shown in Table 1. Thus, it could be seen that the variables were integrated of order 1. Given the order of integration of the series, it becomes pertinent that the long-run state of the system is

**Table 3. VAR lag order selection criteria.**

| Endogenous variables: DPSCP DLEXR DLPDS DLPDM DLNTP DLTEDS | | | | | | |
|---|---|---|---|---|---|---|
| Lag | LogL | LR | FPE | AIC | SC | HQ |
| 0 | 70.55917 | NA* | 3.87e-07* | -3.413291* | -3.242669* | -3.352073* |
| 1 | 83.56569 | 22.67804 | 4.54e-07 | -3.259779 | -2.406671 | -2.953691 |

Source: Authors compilation.

**Table 4. Normalized long run estimates.**

| LPSC | LPDM | LNTP | LTEDS |
|---|---|---|---|
| 1.000000 | -0.928745 | -1.324377 | 0.603401 |
| | (0.05736) | (0.20594) | (0.11684) |
| T-Statistics | [16.19151] | [6.43089] | [5.16434] |

Source: Authors compilation.

established using [44,45] maximum co-integration tests procedure involving trace and maximum eigen statistics as presented in Table 2.

Analysis of the cointegration outcome shows the presence of at least a cointegration for the trace and maximum eigen statistic. This implies that there exists a long run relationship between private sector credit performance and the determinants variables EXR, PDS, PDM, NTP and TEDS. The outcome of the cointegration test suggests that the results of the long run estimates of the private sector credit is reliable and the ability of the study findings to withstand the test of time.

The result of the cointegration result in Table 2 shows the existence of at 3 and 2 cointegrating series for the trace and max-eigen statistic. This therefore, connotes the existence of a long-run relationship between private sector performance and its macroeconomic determinants of exchange rate, public debts servings, public debt management in the model, net taxes on products and total external debt stock.

In determining the long run relationship between private sector credit and its predictors, it essential that the lag structure of the model should be determine. This was examined with Log likelihood ratio (LogL), Likelihood ratio (LR), FPE, Aikaic Information Creteria (AIC), Schwerz Criteria (SC) and HQ. The study however applied lag order of 1 based on SIC, HQ and AIC the minimum statistic (Table 3). The long run estimates of the model are discussed in Table 4.

Table 4 result is functionally expressed in econometric equation after the sign changes as:

$$LPSC = 1.00 + 0.9287LPDM + 1.3244LNTP - 0.6034LTEDS + \mu_t \qquad (4)$$

The result of the normalized cointegration shows the cointegrating coefficients in the long-run relationship between public debt management, net taxes on products, and private sector performance. Considering the normalized cointegrating estimates and the signs interpreted in reverse order, it is seen that there exists a significant positive relationship between debt management net taxes on products and private sector credit. A more explicit interpretation of the result shows that a percentage rise in debt management expenses and net taxes on products accounted for 0.92 and 1.32% rise in private sector credit. This result implies that improved debt management and a rise in net taxes on products are significant factors that enhance private credit availability.

When the government borrows extensively from the private sector to finance its debt, it could lead to crowding out of the private sector investors through rising interest rates and also reduces the available credit. This dissuades the private sector credit mobilization [46]. Also, high levels of public debt can erode investors' confidence in the economy resulting in capital flight and decreased investment. This would consequently have a negative impact on private-sector credit mobilization thereby necessitating the need for an efficient and strategic debt management approach [47,48]. These studies in line with the present study evidence further allude to the far-reaching negative implications of poor debt management on private business sector activities and to the general economy.

**Table 5. Dynamic VEC estimates.**

| DLPSC(-1) | DLPDM(-1) | DLNTP(-1) | DLTEDS(-1) |
|---|---|---|---|
| 1.000000 | -0.488291 | -0.650160 | 0.729498 |
| Std. Error | (0.17470) | (0.32270) | (0.13573) |
| T-Statistics | [-2.79510] | [-2.01472] | [5.37454] |

Source: Authors compilation.

Net taxes on products can reduce consumer's disposable income thereby limiting their purchasing power and decreasing demand for credit. To this effect state actors could consider the implementation of tax reforms that lessens the tax burden on consumers. This will invariably increase their disposable income resulting in greater stimulation for credit demand that will promote private business operations. The result further agrees with an earlier study by [49] who asserts that tax policies affect decision-making processes on work, savings, inter-state migration, investment, and business organization. This is because the private sector generates the funding for the public sector to service the national debt through tax avenues and remittances. Proper debt management will not only help to reduce crowding out of the private sector but could also enhance the global macroeconomic stability and perception of the economy which will further promote private sector credit support and private business partnership with government and foreign investors. High tax returns from private organizations to the government would invariably trigger more financial support from the government to the private sector establishments.

However surprisingly, total external debt stock revealed a significant positive on private sector with an estimate of 0.603. This indicates that a percentage increase in total external debt stock increases private sector credit by 0.603% holding other variables at constant, Thus, the empirical evidence from this study suggests that external debt stock accounted for a significant positive influence on private sector liquidity through credit supply. Notably and more interestingly the past total external debt exacts a positive effect on private sector performance. This suggests that the government could cautiously explore external borrowing to provide additional resources for investment to stimulate private sector activity.

Table 5 is represented in its econometric form as;

$$LPSC = 1.00 + 0.4883LPDM(-1) + 0.6502LNTP(-1) - 0.7295LTEDS(-1) + \mu_t \quad (5)$$

The result of the long run co integrating series is interpreted in its reverse signs after introducing the equation sign. With the series integrated of order, the generated series of the variables incorporated in the model were estimated at their differenced level. The analysis of the result in Table 6 shows that all the determinants of private sector credits exert a significant positive relationship with private sector credit mobilization with the exception of total external debt stocks. Thus, it could be seen that all the variables indicate a significant relationship with

**Table 6. Error correction result.**

| Error Correction: | D(DPSC) | D(DLPDM) | D(DLNTP) | D(DLTEDS) |
|---|---|---|---|---|
| CointEq1 | -0.820547 | 0.170518 | 0.327311 | -0.530813 |
| | (0.14423) | (0.16725) | (0.12194) | (0.22743) |
| | [-5.68911] | [1.01954] | [2.68420] | [-2.33397] |

Source: Authors computation.

**Table 7. Diagnostic tests.**

|  |  | P-value |
|---|---|---|
| R-squared | 0.596039 |  |
| Adj. R-squared | 0.532920 |  |
| Serial Correlation LM Tests | 1.087317 | 0.3809 |
| Heteroskedasticity Tests | 112.7549 | 0.1807 |

Source: Authors computation.

private sector credit. Most essentially total external debts show a significant inverse relationship with private sector credit performance when interpreted in its reverse sign as shown in Eq (5).

Evidence from the dynamic model in equatio4 shows that previous year's management of public debt and total net taxes on products have positively and significantly enhanced private sector credit supply by 0.49 and 0.65%, respectively. This implies that proper management of public debt and revenue generation from taxes are significant factors that helps sustainability and survival of the private enterprises through adequate credit supply that will ensure smooth business operations within the system. Specifically, the analysis of the estimated long run coefficient shows that a percentage increase in past year public external debt retards private sector credit by 0.73%, holding other variables at constant at 1% level of significance. This implies that external public debt service constitutes the most significant factor that negatively affects private sector performance in Nigeria. Given the study is based on a time series data, it pertinent to note that the result is interpreted in the context of Nigeria economy. Hence, future researches could incorporate other African economies using a panel data estimation procedure for a broader generalization of the result based on empirical evidences.

The result in Table 6 shows the short-run adjustment process of the system. The estimated error correction term shows a statistically significant error term, that is correctly signed and within the threshold of zero and 1. The result shows that the system corrects at the rate of 82.1% of the errors encountered as a result of disturbances in the system within one year. Thus, it exhibits a relatively high speed of convergence.

In to ascertain the validity and reliability of the estimated study model, a diagnostic test for the model fitness using the R-squared (0.596) and the adjusted R-squared (0.533) was examined (Table 7). The result however suggests that the independent variables of the model explain 59.60 percent of the total variations in private sector credit mobilization while the adjusted R-squared explains 53.3% of the total variations in the model as a result of the exogenous variables after accounting for model adjustments. This shows the model is of good fit while the serial LM correlation test 1.087; P-value > 0.05) and heteroskedasticity test (112.75; P-value>0.05) attests to the absence of serial autocorrelation of the model estimates with the endogenous variable and the fact that the constant variance assumption of the residual series was not violated in the estimated model of the study. Thus, the study's empirical evidences are credible and valid for sustainable policy recommendations.

## 5. Conclusion and policy recommendations

Sequel the important role played by public debt management and revenue generation through net taxes on products it is important that the government derive a more innovative and proactive approach to public debt management and internally generated revenue drive rather than relying on external sources of finance in fiscal financing of projects and budgets. This will significantly improve the international reputation of the economy and will not internally or

externally not crowd out the private sector in credit mobilization. Adequate credit and capital for the private sector establishment will not only promote their growth but will further guarantee their sustainability and efficiency in its holistic operations. It is therefore imperative for government stakeholders to pay attention to the management of national debt and its impact on the private sector.

Hence when the nation's debt profile is not adequately managed it could create a wrong notion of the financial stability of any economy. This would also the international reputation and financial credibility of such an economy which could have a resultant negative effect on the capital inflows to the private sector of the economy. This study stresses the need to exercise high caution in the Nation's debt management approach and for state actors to always sustain policies that will mitigate against financial prudence in national borrowing and spending with its attendant adverse consequences on the private sector performance. A high debt profile could send a wrong signal to business investors hence, policymakers should endeavor to concentrate on the promotion of transparency, strengthening governance, and strive to implement sound debt management practices to restore and enhance investors' confidence. Policymakers could consider the implementation of measures to reduce the crowding-out effect, such as improving fiscal discipline and exploring other sources of project financing.

Therefore, policymakers should ensure that external borrowing is sustainable and does not lead to excessive debt burdens which entails careful management of borrowing terms, repayment schedules, and interest rates. However, is pertinent to note here that the study scope is country-specific by nature and likewise it results and interpretations. Thus, further research could expand the scope by considering more economies across Africa and beyond. In that case, it would be possible to know whether a common economic trend could be established using a panel data analytical process. Future research studies could also consider the possible inclusion of theoretical constructs to account for economic policy uncertainty which could be simulated under the model calibration approach.

## Supporting information

**S1 Data.**
(XLSX)

## Author Contributions

**Conceptualization:** Mengya Shang, Yin Hang, Daniel E. Ufua.

**Data curation:** Uchechukwu E. Okorie.

**Formal analysis:** Uchechukwu E. Okorie.

**Software:** Uchechukwu E. Okorie.

**Supervision:** Xiaosong Jin.

**Visualization:** Daniel E. Ufua.

**Writing – original draft:** Daniel E. Ufua.

**Writing – review & editing:** Xiaosong Jin, Daniel E. Ufua.

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
