## [Decision Letter · Decision Letter 0]

11 Sep 2023

PONE-D-23-24997National Debt Management and Business Sustainability in Africa’s Largest Economy: A Focus on the Private SectorPLOS ONE

Dear Dr. UFUA,

Thank you for submitting your manuscript to PLOS ONE. After careful consideration, we feel that it has merit but does not fully meet PLOS ONE’s publication criteria as it currently stands. Therefore, we invite you to submit a revised version of the manuscript that addresses the points raised during the review process.

We look forward to receiving your revised manuscript.

Kind regards,

Magdalena Radulescu

Academic Editor

PLOS ONE

Journal Requirements:

 "Not Applicable"

4. Please ensure that you include a title page within your main document. You should list all authors and all affiliations as per our author instructions and clearly indicate the corresponding author.

5. Please upload a new copy of Figure 1 as the detail is not clear. Please follow the link for more information: https://blogs.plos.org/plos/2019/06/looking-good-tips-for-creating-your-plos-figures-graphics/
https://blogs.plos.org/plos/2019/06/looking-good-tips-for-creating-your-plos-figures-graphics/

6. Please include a copy of Table 4.1 and table 4.7 which you refer to in your text on page 6 and 8.

7. We note you have included a table to which you do not refer in the text of your manuscript. Please ensure that you refer to Table 4 and table 4.8 in your text; if accepted, production will need this reference to link the reader to the Table.

**Additional Editor Comments:**

Please check reviewers'reports. Elaborate a response letter to show how you have addressed to their comments and mark in color all changes you will make into the manuscript.

Reviewers' comments:

Reviewer's Responses to Questions

**Comments to the Author**

1. Is the manuscript technically sound, and do the data support the conclusions?

Reviewer #1: Yes

Reviewer #2: No

2. Has the statistical analysis been performed appropriately and rigorously? 

Reviewer #1: No

Reviewer #2: Yes

3. Have the authors made all data underlying the findings in their manuscript fully available?

Reviewer #1: No

Reviewer #2: Yes

4. Is the manuscript presented in an intelligible fashion and written in standard English?

Reviewer #1: No

Reviewer #2: No

5. Review Comments to the Author

Reviewer #1: Thank you very much for inviting me to assess the above-mentioned manuscript submitted to Plos One. From the research topic, this paper attempts to examine “National Debt Management and Business Sustainability in Africa’s Largest Economy: A Focus on the Private Sector” approaches. After reading this paper, I have too many comments.

Firstly I suggest authors to rewrite the abstract to make it more constructive. Abstract should have at least one sentence per each: context and background, motivation, hypothesis, methods, results, conclusions. Add some numbers from your findings and take out paranthesis from the abstract

i) The introduction part of the study needs improvement and story flow and the authors need to give proper contributions to their study. İi) It is better to affard the contributions of the paper in the introduction part. İii) I would like to suggest that authors should update the literature for intro and all text.

There is a need to do a more rigorous and systematic literature review. The authors should clearly mention the literature gap. See following literatures and please kindly add them https://doi.org/10.1080/15567249.2016.1263251;
https://doi.org/10.1007/s11356-023-28977-w;
https://doi.org/10.1002/jtr.2151;
https://doi.org/10.1007/s11356-019-04514-6;
https://doi.org/10.1007/s11356-021-16720-2;
https://doi.org/10.1007/s13132-011-0075-2;
https://doi.org/10.1007/s12667-010-0018-1;
https://doi.org/10.1007/s11356-021-12993-9;
https://doi.org/10.1007/s11356-019-06276-7;
https://doi.org/10.1007/s11356-021-12637-y;
https://doi.org/10.1016/j.resourpol.2023.103300;
https://doi.org/10.1177/13548166231174812

It would be appropriate to indicate a sharp future research directions and limitations of this at the end of the conclusion section just before references. Need clear future recommendation/implementation in the context of uncertainty. See and add kindly followings to your text: https://doi.org/10.1177/1354816619888346;
https://doi.org/10.1007/s00477-023-02452-x

Language quality: The paper needs an extensive language editing with the help of a professional language editor to improve its quality.

Reviewer #2: I am writing about the manuscript (PONE-D-23-24997) entitled “National Debt Management and Business Sustainability in Africa’s Largest Economy: A Focus on the Private Sector” Thank you for the opportunity to review this paper. The paper is not well-structured and conveys a deal of information. I recommend for the paper substantial modifications and refinements of the present version. My comments are as follows:

1. The paper uses complex language and technical terms, potentially challenging readers without relevant backgrounds. A revision is needed to make the abstract more accessible and professionally written.

2. The study's scope in addressing knowledge gaps appears limited; a broader examination of existing gaps could enhance the research's significance.

3. No clear justification is provided for the selection of specific variables in the study, which should be clarified to enhance the research's validity.

4. the theoretical framework is not well-explained, the study's specific contributions to addressing literature limitations and gaps are not adequately discussed.

5. The relationship between the selected variables in the results and the theoretical framework should be justify by valid theoretical framework.

6. The methods section lacks explicit justification for the chosen study design and methods, as well as the management of potential biases, necessitating additional clarification.

9. The results section does not clearly explain how the results are organized, how they relate to the research question, or how they will inform the study.

10. The discussion section does not provide a clear comparison and contrast between the study's findings and relevant literature in the field, nor does it explain unexpected findings.

11. The discussion section lacks a clear explanation of how study limitations may have affected the results, how the results will be used to inform future research or practice, or a clear and concise conclusion.

12. The presentation of tables and figures is not always consistent with the text description, and the figure legends and table headings do not always clearly explain the content.

13. The references do not always include up-to-date sources, and discrepancies exist between citations and the reference list.

14. In the conclusion section, there is a lack of further insights or suggestions for future research, and limitations are not always acknowledged. Additionally, critical analysis of policy recommendations is limited.

15. The existing literature requires updating, and a comprehensive summary should be provided at the end of this section. The summary must highlight areas of agreement and divergence among previous studies, pinpoint aspects they did not address, and establish connections to the objectives of the current study.

16. The discussion section needs more previous studies, studies, and examples of agree and disagree points to support the current results.

17. The study needs to incorporate the limitations and future direction and future direction as well.

6. PLOS authors have the option to publish the peer review history of their article (what does this mean?). If published, this will include your full peer review and any attached files.

Reviewer #1: No

Reviewer #2: No

---

## [Author Response · Author response to Decision Letter 0]

4 Oct 2023

The authors have addressed this comment. See page 1

We have uploaded figure 1 in a separate file

The Tables being referenced to in Pages 6 and 8 have been correctly labeled in referenced accordingly 

All Tables and have included and referred to in the text.

We have renamed the tables, following serial order

Authors ensure these are duly addressed in the revision.

The abstract was constructively written to include context and background, motivation, hypothesis, methods, results and conclusion. 

This comment has been addressed. See Page 3

The authors thank the Reviewer for this comment and the literature. We have engaged them to improve on the manuscript

Future research directions and limitations were incorporated at the end of the conclusion section. 

Future recommendation in the context of uncertainty was also added.

The authors engaged in a thorough revision of the manuscript to address this comment.

The authors have revised the abstract and the entire manuscript.

More explanation based on extant literature have been provided both in the introduction and the literature review sections

The authors have provided a clearer explanation of this.

An underpinning theory has been added to the manuscript.

The study is hinged on the schematic model of national debt management (Figure1) and the system theory, see page 7

The relationship between the selected variables in the result and the schematic and theoretical frame work was justified. See page 8, highlighted in red ink

Explicit justification for the chosen design and methods, as well as management of potential bias were further clarified in the methodology section. See page 8

The organization of the results was included in this section and how the results were linked with the research question. 

The discussion section provided a comparison with relevant literature in field. It also explained unexpected findings

The explanation of how the study limitation could have affected the result and how it would inform future research was added in this section. 

The tables and figures have been aligned with text description and the figure legends and Table headings clearly explained 

Authors have addressed this comment. More literature has been added. See literature section and references, highlighted in red ink

Limitations, further insights and suggestions for future research were added to the concluding part of the study. See highlighted in red

Limitations, further insights and suggestions for future research were added to the concluding part of the study. Critical analysis of policy recommendations was included. 

Authors have addressed this comment throughout the manuscript. More recent literature has been added to improve the work.

More previous studies were added with examples of agree points to support the current results.

The limitations and future direction of study were incorporated in the study

---

## [Decision Letter · Decision Letter 1]

17 Oct 2023

National Debt Management and Business Sustainability in Africa’s Largest Economy: A Focus on the Private Sector

PONE-D-23-24997R1

Dear Dr. UFUA,

We’re pleased to inform you that your manuscript has been judged scientifically suitable for publication and will be formally accepted for publication once it meets all outstanding technical requirements.

Kind regards,

Magdalena Radulescu

Academic Editor

PLOS ONE

Additional Editor Comments (optional):

Reviewers' comments:

Reviewer's Responses to Questions

**Comments to the Author**

1. If the authors have adequately addressed your comments raised in a previous round of review and you feel that this manuscript is now acceptable for publication, you may indicate that here to bypass the “Comments to the Author” section, enter your conflict of interest statement in the “Confidential to Editor” section, and submit your "Accept" recommendation.

Reviewer #1: All comments have been addressed

Reviewer #2: All comments have been addressed

2. Is the manuscript technically sound, and do the data support the conclusions?

Reviewer #1: Yes

Reviewer #2: Yes

3. Has the statistical analysis been performed appropriately and rigorously? 

Reviewer #1: Yes

Reviewer #2: Yes

4. Have the authors made all data underlying the findings in their manuscript fully available?

Reviewer #1: Yes

Reviewer #2: (No Response)

5. Is the manuscript presented in an intelligible fashion and written in standard English?

Reviewer #1: Yes

Reviewer #2: (No Response)

6. Review Comments to the Author

Reviewer #1: Thank you for giving me the opportunity to read your paper. The paper “PONE-D-23-24997” is interesting for journal readers. But following changes should be done before the consideration to improve the quality of the paper:

Reviewer #2: (No Response)

7. PLOS authors have the option to publish the peer review history of their article (what does this mean?). If published, this will include your full peer review and any attached files.

Reviewer #1: No

Reviewer #2: No

---

## [Editor Report · Acceptance letter]

23 Oct 2023

PONE-D-23-24997R1 

National Debt Management and Business Sustainability in Africa’s Largest Economy: A Focus on the Private Sector 

Dear Dr. Ufua:

I'm pleased to inform you that your manuscript has been deemed suitable for publication in PLOS ONE. Congratulations! Your manuscript is now with our production department. 

Kind regards, 

on behalf of

Dr. Magdalena Radulescu 

Academic Editor

PLOS ONE